# Glucosinolate Biosynthetic Genes of Cabbage: Genome-Wide Identification, Evolution, and Expression Analysis

**DOI:** 10.3390/genes14020476

**Published:** 2023-02-13

**Authors:** Peng Wang, Wenxue Cao, Limei Yang, Yangyong Zhang, Zhiyuan Fang, Mu Zhuang, Honghao Lv, Yong Wang, Shanhan Cheng, Jialei Ji

**Affiliations:** 1Key Laboratory for Quality Regulation of Tropical Horticultural Crops of Hainan Province, College of Horticulture, Hainan University, Haikou 570228, China; 2Institute of Vegetables and Flowers, Chinese Academy of Agricultural Sciences, Beijing 100081, China; 3College of Horticulture and Landscape Architecture, Hunan Agricultural University, 1 Nongda Road, Changsha 410128, China

**Keywords:** glucosinolate biosynthetic gene, cabbage, exogenous hormone treatment, gene expression, phylogeny

## Abstract

Cabbage (*Brassica oleracea* var. *capitata*) is a vegetable rich in glucosinolates (GSLs) that have proven health benefits. To gain insights into the synthesis of GSLs in cabbage, we systematically analyzed GSLs biosynthetic genes (GBGs) in the entire cabbage genome. In total, 193 cabbage GBGs were identified, which were homologous to 106 GBGs in *Arabidopsis thaliana*. Most GBGs in cabbage have undergone negative selection. Many homologous GBGs in cabbage and Chinese cabbage differed in expression patterns indicating the unique functions of these homologous GBGs. Spraying five exogenous hormones significantly altered expression levels of GBGs in cabbage. For example, MeJA significantly upregulated side chain extension genes *BoIPMILSU1-1* and *BoBCAT-3-1,* and the expression of core structure construction genes *BoCYP83A1* and *BoST5C-1*, while ETH significantly repressed the expression of side chain extension genes such as *BoIPMILSU1-1*, *BoCYP79B2-1,* and *BoMAMI-1*, and some transcription factors, namely *BoMYB28-1*, *BoMYB34-1*, *BoMYB76-1*, *BoCYP79B2-1,* and *BoMAMI-1*. Phylogenetically, the *CYP83* family and *CYP79B* and *CYP79F* subfamilies may only be involved in GSL synthesis in cruciferous plants. Our unprecedented identification and analysis of GBGs in cabbage at the genome-wide level lays a foundation for the regulation of GSLs synthesis through gene editing and overexpression.

## 1. Introduction

Glucosinolates (GSLs) are an important class of plant secondary metabolites. Currently, about 200 GSLs are known, which are distributed in 16 families of dicotyledonous plants, especially in *Brassicaceae* [1,2,3]. The GSL hydrolysis products display diverse bioactivities, function both in defence and as an attractant in plants, play a role in cancer prevention in humans, and act as flavour compounds. Due to their effects on both plants and humans, GSLs are a current biology research focus [4,5,6,7,8,9]. Indeed, rapeseed breeders focus on harmful GSLs in rapeseed cakes, vegetable breeders are more interested in anticancer GSLs that are beneficial to humans, and medical scientists focus on GSLs that are inhibitory to tumor cells [10,11,12,13,14]. Sulforaphane, a secondary metabolite, relieves neuropathic pain caused by chemotherapy and inhibits prostate, colon, breast, pancreatic, and bladder cancers [15,16,17,18,19].

Structurally, all GSLs comprise a β-D-glucopyranose residue linked through a sulfur atom to a (Z)-N-hydroximinosulfate ester plus an R side-chain group derived from an amino acid (AA). Based on different AA sources of R groups, GSLs can be divided into three categories: aliphatic GSLs (R side chains are derived from methionine, alanine, valine, leucine, and isoleucine), aromatic GSLs (R side chains are derived from amino acids containing aromatic rings, tyrosine, and phenylalanine), and indole GSLs (R side chains are derived from tryptophan). The GSL synthetic pathway has been thoroughly studied in the model plant *Arabidopsis thaliana* [20,21,22,23,24,25,26,27,28,29,30,31,32,33,34,35,36,37,38]. The biosynthesis of glucosinolates can be summarized into three stages: extension of the R side chain, synthesis of the core structure, and modification of the side chain [39] (Figure 1) Figure 1 is quoted from references [40,41].

Cabbage (*Brassica oleracea* L. var. *capitata* L.) is an important cruciferous vegetable that is rich in GSLs and widely grown globally. Although the whole-genome sequence of cabbage is available, there is a paucity of studies on cabbage GSL biosynthetic genes (GBGs). Therefore, to gain an in-depth understanding of the regulatory mechanism of GSL synthesis in cabbage, we conducted genome-wide identification, evolution, and expression analyses of cabbage GBGs using the *Arabidopsis* GSL synthesis pathway as a reference. This study establishes a reference for improving the composition and content of cabbage GSLs through molecular breeding technologies.

## 2. Materials and Methods

### 2.1. Data Sources

The sequence and annotation data of *Arabidopsis* GBGs were retrieved from the TAIR database (https://www.arabidopsis.org (accessed on 18 May 2022)). The genomic and annotation data of both cabbage and Chinese cabbage (*B. rapa* pekinensis) were retrieved from the BRAD database (http://brassicadb.cn (accessed on 21 May 2022)).

### 2.2. Identification of GBGs in Cabbage

To identify cabbage GBGs, we queried the cabbage proteins in the BRAD database with the protein sequences of *Arabidopsis* GBGs using BLASTP with *E*-value ≤ 10^−10^ and coverage ≥ 0.75 cutoffs. Syntenic orthologous GBGs between *Arabidopsis* and cabbage were identified according to the sequence similarity (*E*-value ≤ 10^−20^) of flanking genes. The specific distribution of cabbage GBGs on chromosomes was analyzed using MG2C (http://mg2c.iask (accessed on 5 June 2022)).

### 2.3. Non-Synonymous/Synonymous Substitution (Ka/Ks) Analysis

To evaluate the selection pattern of GBGs between *Arabidopsis* and both cabbage and Chinese cabbage, we used MEGA 7 software [42] to calculate the value of Ka/Ks of the orthologous gene pairs. A value of Ka/Ks greater than one represents positive selection, a value of Ka/Ks equal to one represents neutral selection, and a value of Ka/Ks less than one represents negative selection.

### 2.4. Expression Analysis of GBGs

The expression levels of GBGs in seven different organs (root, callus, leaf, stem, bud, flower, and silique) of cabbage and Chinese cabbage were determined using RNA-seq data in the Gene Expression Omnibus database under accession number GSE42891 and GSE43245, respectively.

To investigate the effect of different phytohormones on the expression of GBGs in cabbage, we sprayed leaves with five exogenous phytohormones: ethylene (ETH), abscisic acid (ABA), salicylic acid (SA), strigolactone (SL), and methyl jasmonate (MeJA) at concentrations of 100 mg/L, 50 mg/L, 100 µM, 5 µM, and 100 µM, respectively. As a negative control, a set of leaves were sprayed with double distilled water. The control group and five treatment groups were sprinkled with 200 mL solution with a spray kettle. Leaves were sampled at 2, 4, 6, 12, and 24 h after spraying. The cabbage were grown in a climate-controlled greenhouse with 20/15 °C (12/12 h) day/night temperature and 70% relative humidity for 20–30 days.

Six tissues, including the root, leaf, stem, flower, callus, and silique of *B. oleracea* accession ‘02-12’ were used for RNA extraction. Samples were rapidly obtained and placed in liquid nitrogen, and all collected samples were stored in a cryogenic freezer at −80 °C. RNA was extracted using the RNAprep Pure Plant Kit. Residual genomic DNA was removed with DNase 1, RNase-free and the quality of RNA was detected by agarose gel electrophoresis. The cDNA was synthesized using a FastKing RT Kit (Tiangen Biotech CO., Ltd., Beijing, China) and the reaction conditions were as follows 25 °C for 5 min, 37 °C for 45 min, 85 °C for 5 s. The relative expressions of glucosinolate genes were determined using qRT-PCR. qRT-PCR was performed using the ChamQ™ Universal SYBR^®^ qPCR Master Mix (Vazyme Biotech CO., LTD, Nanjing, China) on a Bio-Rad CFX96TM Real-Time System (Bio-Rad, Hercules, CA, USA). Each reaction has a total volume of 10 μL, which includes 10 ng cDNA, 0.01 nmol forward and reverse primers, 5 μL of SYBR qPCR Master Mix, and 0.2 μL of 50 × ROX reference dye. Predenaturation was carried out at 95 °C for 30 s, 40 cycle reactions: 95 °C for 10 s, 60 °C for 30 s, and melting curve: 95 °C for 15 s, 60 °C for 1 min, 95 °C for 15 s. Relative expression was calculated using the 2^−ΔΔCT^ method. All reactions for qPCR were performed with three biological and technical replicates.

### 2.5. Phylogenetic Analysis

P450 amino acid sequences from seven plant species were obtained from the cytochrome P450 database (https://drnelson.uthsc.edu/plants/ (accessed on 16 June 2022)). These protein sequences were aligned by MAFFT (v7.037) [43] with default parameters. The phylogenetic tree was constructed using FastTree with WAG + CAT model [44].

## 3. Results

### 3.1. Identification of Glucosinolate Biosynthetic Genes in B. oleracea

Glucosinolate biosynthetic genes from *A. thaliana* (AtGBSs) were used to identify *B. oleracea* glucosinolate biosynthetic genes (BoGBGs). A total of 193 BoGBGs were identified (Appendix A), of which 169 BoGBGs were syntenic orthologs of the 106 AtGBGs (Figure 2). Based on the genome annotation file, 169 BoGBGs were mapped to nine chromosomes. As shown in Figure 3, there were 18, 25, 26, 15, 17, 16, 18, 14, and 20 BoGBGs located on chromosomes C01-C09, respectively.

### 3.2. Comparative Evolutionary Analyses of Orthologous Gene Pairs of Glucosinolate Biosynthetic Genes

To explore the selection pressure acting on GBGs in *B. oleracea*, *A. thaliana* and *B. rapa,* Ka/Ks values for BoGBGs, BrGBGs and AtGBGs were comparatively analyzed. Ultimately, 20 homologous gene pairs of GBGs were selected (Figure 4). The Ka/Ks values of GBGs in *B. oleracea*, *B. rapa,* and *A*. *thaliana* were similar. However, the Ka/Ks ratios of *BoMAM1-1*, *BoST5a-1*, *BoSUR1-1,* and *BoAOP1-1* were slightly lower in *B. oleracea* than in *B. rapa* genomes. Except for *BoST5a*, Ka/Ks values were all less than one, indicating that most genes were under negative selection (Figure 4).

### 3.3. Expression Analysis of Glucosinolate Biosynthetic Genes Orthologs from B. oleracea, B. rapa, and A. thaliana genomes

The expression levels of BoGBGs and BrGBGs in different tissues (callus, flower, leaf, root, silique, and stem) were analyzed based on a publicly available transcriptomic dataset. The transcript levels of 133 glucosinolate biosynthetic genes were assessed (Appendix A). From the heatmap visualization, 20 BoGBGs and BrGBGs, which were important for glucosinolate synthesis, were expressed (Figure 5). Seven of these genes had the highest and lowest expression levels in the siliques and roots, respectively: *BoUGT79B1-1*, *BoFMOG5-1*, *BoSUPR1*, *BoBCAT-4-1*, *BoMAM1-1*, *BoCYP83A1-1,* and *BoCYP79F1*. By contrast, the expression levels of *BoAOP1-1*, *BoPYK10-,1* and *BoTBS-1* were highest in the roots and lowest in the siliques. Some genes were highly expressed in cabbage leaves, such as *BoBCAT-3-1*, *BoCYP79A2-1*, *BoCYP79B2-1*, *BoST5a-1,* and *BoGSTF9-1. BoCYP81F2-1*, *BoGMT1-1,* and *BoGSH2-1* were barely expressed, except in the callus. *BoST5a-1* had the lowest and highest expression levels in the stems and leaves, respectively. *BoST5a-2* was expressed in all tissues and had the highest and lowest expression levels in the roots and flowers, respectively. *BoAOP1-1* had the highest expression levels in the roots and stems.

The different expression patterns of orthologous gene pairs indicate possible functional differentiation. We found that *BoMAM1-1* displayed the highest expression levels in the siliques and stems, whereas *BrMAM1-1* displayed the highest expression levels in the roots. *BoCYP79A2-1* was highly expressed in the leaves, whereas the expression levels of *BoCYP79A2-2* and *BoCYP79A2-3* were the highest in the roots. However, *BrCYP79A2-1* was highly expressed in the stems and *BrCYP79A2-2* and *BrCYP79A2-3* were highly expressed in the flowers. Our results showed that some homologous genes in *B. oleracea* and *B. rapa* have different expression patterns, indicating that the functions of these genes possibly diverged with speciation.

The tissue expression of important genes in the glucosinolate biosynthetic pathway was analyzed to further understand the expression of GBGs in *B. oleracea* (Figure 6). The results showed that *BoST5a-1* and *BoST5a-2* were expressed in all organs, including the calluses, flowers, leaves, roots, siliques, and stems, while *BoST5a-3* was only expressed in the flowers. The expression of *BoCYP79A2-1* was highest in the leaves, calluses, and siliques, followed by the stems, flowers, and roots. Almost no expression of *BoCYP79A2-1* was detected in the roots. Meanwhile, the expression of *BoCYP79A2-2* was only expressed in the roots. *BoCYP79A2-3* was expressed in the flowers and roots. *BoMAM1-1* and *BoMAM1-3* were expressed in all tissues. The expression of *BoAOP1-1* was in the roots, stems, and calluses, and no expression or weak expression was observed in other tissues. Gene expression analysis revealed an extensive variance between paralogs of each GBGs.

### 3.4. Twelve Glucosinolate Biosynthetic Genes Respond to Exogenous Phytohormone Treatments

To investigate the effects of different exogenous phytohormones on GBG expression, we sprayed cabbages with ETH, ABA, SA, MeJA, and SL (Figure 7).

At 100 mg/L, ETH promoted the expression of two genes (*BoCYP83A1* and *BoMAM1-1*) 2 h after treatment. The expression of *BoBCAT-3-1* decreased 4 h after treatment. The expression levels of the remaining genes (*BoIPMILSU1-1*, *BoSUPR1-1*, *BoST5C-1*, *BoCYP79A2-1*, *BoCYP79B2-1*, *BoMYB34-1*, *BoMYB28-1*, *BoMYB76-1,* and *BoCYP79F1-1*) were not affected.

At 50 mg/L, ABA promoted the expression of 10 genes (*BoIPMILSU1-1*, *BoBCAT-3-1*, *BoMYB28-1*, *BoCYP79B2-1*, *BoMYB34-1*, *BoMYB76-1*, *BoCYP83A1*, *BoST5C-1*, *BoMAM1-1* and *BoSUPR1-1*). Two other genes (*BoCYP79A2-1*, *BoCYP79F1-1*) were significantly repressed within 24 h of treatment.

Treatment with 100 µmol/L of SA promoted the expression of five genes (*BoIPMILSU1-1-1*, *BoBCAT-3-1*, *BoMAM1-1*, *BoMYB34-1*, and *BoSUPR1-1*). Specifically, the expression of *BoIPMILSU1-1-1* and *BoBCAT-3-1* was promoted 4 h after treatment and that of *BoMAM1* and *BoMYB34-1* 12 h after treatment. The expression of *BoSUPR1-1* was promoted between 4 h to 12 h after treatment. Six genes were completely repressed: *BoCYP79A2-1*, *BoCYP79F1-1*, *BoMYB28-1*, *BoCYP79B2-1*, *BoMYB76-1,* and *BoST5C-1. BoCYP83A1* was promoted up to 4 h after treatment and repressed thereafter.

The expression of five genes (*BoCYP79A2-1*, *BoCYP79B2-1*, *BoMYB34-1*, *BoMYB76-1,* and *BoCYP79F1-1*) was repressed by treatment with MeJA at 100 µmol/L. Two genes (*BoIPMILSU1-1* and *BoMYB28-1*) were initially promoted and then repressed. The expression of *BoST5C-1* was not promoted within 24 h of treatment.

Treatment with 5 µmol/L SL promoted the expression of *BoMAM1-1* and *BoCYP83A1* 12 h and 2 h after treatment, respectively. The expression levels of other selected genes were unaffected by SL treatments.

### 3.5. Evolutionary History of P450 Genes among Seven Plant Species

The cytochrome P450 gene family is important for plant development as it is largely involved in the synthesis and metabolism of plant hormones, pigments, and fatty acids. In this study, P450 genes were identified in *B. oleracea* and six other representative plant species. As shown in Figure 8, 48 genes clustered into four clades (*CYP78*, *CYP79*, *CYP81,* and *CYP83*). *BoCYP83A1* and *BoCYP83B1* clustered with the *CYP83* family (blue area), whereas *BoCYP79F1*, *BoCYP79A2*, *BoCYP79B3*, *BoCYP79B2-1,* and *BoCYP79B2-1* clustered with the *CYP79* family (yellow, green, and pink area).

## 4. Discussion

### 4.1. Characterization of Glucosinolate Biosynthetic Genes in B. oleracea

Whole genome duplication provides genetic material for the expansion of gene families and the evolution of new genes [45]. Three rounds of whole genome duplication occurred in the evolution of *B. oleracea*. We identified 193 GBGs in *B. oleracea,* suggesting their possible evolution from a duplicated gene. A total of 169 GBGs were syntenic orthologs of *A. thaliana*, whereas 24 GBGs had no syntenic relationships. However, 19 AtGBGs (for example, *AT3G44320*, *AT3G16390,* and *AT5G40360*) had no *B. oleracea* orthologs, suggesting their possible loss during the evolution of *B. oleracea*.

To analyze the evolutionary relationships of glucosinolate biosynthesis homologous gene pairs among *A. thaliana*, *B. oleracea,* and *B. rapa,* we compared the non-synonymous/synonymous (Ka/Ks) ratios of GBGs (Figure 4). The Ka/Ks ratio expresses the selective pressure on the evolution of genes [46]. Two genes (*BoST5a* and *BoAOP1*) significantly differed in orthologous gene pairs between *A. thaliana–B. oleracea* and *A. thaliana–B. rapa.* The KA/KS ratio of *BoST5a* in *B. rapa* was above one, indicating that it had undergone positive selection. Conversely, the KA/KS ratio of *BoST5a* in *B. oleracea* was below one, indicating that it underwent negative selection during evolution. For both *BoAOP1* and *BrAOP1*, the ratio was below one. However, more negative selection occurred in *B. rapa*. The Ka/Ks ratios of *BoGUT74B1*, *BoFMOGS-OX5,* and *BoGTM1* were all below one, there are more positive selection occurred in *B. oleracea* compared to *B. rapa*. The remaining homologous genes such as *BoBACT-3* and *BrBACT-3*, *BoCYP83A1,* and *BrCYP83A1* are closely underwent negative selection during evolution. Therefore, we speculated that BoGBGs and BrGBGs underwent different selective pressures.

### 4.2. Exogenous Plant Hormones Regulate the Expression of Glucosinolate Biosynthetic Genes

The biosynthesis of glucosinolate in *B. oleracea* is a complex process regulated by multiple genes and transcription factors. Relatedly, exogenous plant hormones affect the accumulation of glucosinolate. In this study, different concentrations of five exogenous phytohormones (ETH, ABA, MeJA, SA, and SL) were sprayed on cabbage leaves to investigate their effects on the expression of GBGs.

In *A. thaliana*, the *MAM* locus has three duplicate genes, including *MAM1*, *MAM2,* and *MAM3*. They are involved in the elongation of the R chain during the biosynthesis of GSLs [47]. For example, *MAM2* catalyzes the first elongation only, whereas *MAM1* and *MAM3* catalyze C3-C5 and C3-C8 reactions, respectively [48]. The expression of *BoMAM1-1* in cabbage leaves significantly increased after ETH and ABA treatments. Therefore, the application of ETH and ABA may be beneficial to the synthesis of short-chain aliphatic thiosinolates. Moreover, overexpression of the *MAM1* gene in *Brassica napus* significantly increased the content of aliphatic thiosinolates [49]. We speculate that spraying cabbage leaves with ETH and ABA increases aliphatic thiosinolates. The next step is the production of elongated 2-oxo acid catalyzed by isopropionate isomerases, which can participate in the core structure of GSL synthesis. *IPMI,* one of isomerases, can create chain-elongated derivatives of methionine such as dihomomethionine, trihomomethionine, tetrahomomethionine, pentahomomethionine, and hexahomomethionine. In *Arabidopsis thaliana*, isopropylmate isomerases (IPMI) consists of a single large (*IPMI LSU1*) and one of three different small subunits (*IPMI SSU1* to *SSU3*). *IPM1* and transcription factor *MYB28* are co-expressed to positively regulate the synthesis of GLS. MeJA significantly upregulated the expression of *BoIPMILSU1-1*. Thus, it may promote amino acid side chain to extension in cabbage. Interestingly, MeJA inhibits the expression of *BoMYB28*. We speculate that MeJA is more beneficial in promoting the synthesis of long-chain methionine. The next side chain extension process is the transamination process involving *BCAT3*, *BCAT4,* or *BCAT6*. The chain-lengthened 2-oxoacid is catalyzed to the corresponding amino acid while the production of homomethionine from methionine. ETH, ABA, and MeJA can increase the expression level of *BoBCAT3-1.* This may promote the synthesis of aliphatic thioglycosides.

Glucosinolate biosynthesis is a complex process regulated by many genes and transcription factors. Cytochrome P450 from *CYP79* and *CYP83* families are involved in synthesizing the core structure of glucosinolates. *AtCYP83A1* converts the aliphonic aldehyde group to an activated acylnitro compound. Indeed, the lipid content in the *CYP83A1* deletion mutant in *Arabidopsis* was significantly lower than that of the wild type [50,51,52]. Spraying cabbage leaves with MeJA, ETH, and ABA upregulated *BoCYP83A1* expression. Therefore, we speculate that spraying these three hormones increases the content of aliphatic thiosinolates in cabbage leaves. CYP79B2/B3, important enzymes in the glucoside biosynthesis pathway, convert tryptophan to indole-3-acetylaldoxime (a precursor of Indole-3-aceticacid). The expression level of *CYP79B2-1* in *B. oleracea* increased shortly after the application of SL, which possibly temporarily increased indole-glucosinolide and auxin content in cabbage leaves. *CYP79F1* is involved in the biosynthesis of short and long-chain aliphatic glucosinolates, while *CYP79F2* only catalyzes the production of long-chain aliphatic glucosinolates [53,54,55]. In *CYP79F1* knockout mutants of *A. thaliana*, the content of short-chain aliphatic glucosinolates significantly decreased while levels of auxins and cytokinins increased. After spraying cabbage leaves with ABA, *BoCYP79F1-1* expression significantly decreased and this repression decreased with time. Spraying cabbage leaves with ABA possibly reduced their content of short-chain aliphatic glucosinolates, auxins, and cytokinins.

*MYB34* is an important transcription factor of indole glucosinolate biosynthesis. In *A. thaliana*, *MYB34* positively regulates the transcription of *CYP79B2/B3* genes for the biosynthesis of indole glucosinolates [56,57]. Additionally, in *Arabidopsis*, the expression of *AtMYB34* is inhibited through ABA and MejA signaling pathways, thereby negatively regulating the synthesis of glucoside. The expression level of *BoMYB34* significantly decreased in cabbage leaves within 24 h after ABA and MejA treatments, which was similar to *Arabidopsis*. Therefore, *BoMYB34-1* had a positive regulatory effect on indoles glucosinolate biosynthesis in cabbage. *MYB28* and *MYB76* transcription factors were highly associated and co-expressed with structural aliphatic glucosinolates biosynthetic genes, such as *BCAT4*, *MAM1*, *CYP79F2*, and *CYP83A1*. Essentially, *MYB28* and *MYB76* are closely related to the regulation of aliphatic thioside biosynthesis. The aliphatic thioside content in MYB28-overexpressed plants was altered compared to wild-type *Arabidopsis*. Metabolic analysis showed 3C-5C glucosinolates were increased significantly. However, the content of long-chain aliphatic glucosinolates remained unchanged or decreased slightly. Practically, while ABA promotes *MYB28* expression, the expression levels of *BCAT4*, *MAM1*, *CYP79F2,* and *CYP83A1* are also significantly upregulated. It is well illustrated that *BoMYB28* may have a co-expression relationship with *BoBCAT4*, *BoMAM1*, *BoCYP79F2,* and *BoCYP83A1* in cabbage. Compared with wild-type *Arabidopsis*, levels of both short-chain and long-chain aliphatic glucosinolates in *MYB76* overexpressed plants were increased. However, the promotion effect of *MYB76* is weak compared to the *MYB28* regulator. For example, the content of 4C glucosinolates is 3–4 times higher than that of wild type, and the transcription level of *MYB76* needs to be increased by 100 times, while the transcription level of *MYB28* only needs to be increased by 14 times. Interestingly, overexpression of *MYB76* not only leads to the accumulation of aliphatic glucosinolates, but also to an increase in the content of indole glucosinolates. After spraying ABA, the expression of *MYB76* was significantly inhibited within 12 h. Therefore, it might be detrimental to the biosynthesis of indole glucosinolates in cabbage leaves.

In the future, gene editing will be used to verify the biological function of some significant *BoGBGs* in cabbage. Thus, the materials with a high content of the beneficial thioglycoside glucosinolates are obtained.

### 4.3. Role of CYP450 in Glucosinolate Biosynthesis in Members of Cruciferae

Plant cytochrome P450 is an important oxygenating enzyme in the synthesis of glucosinolates in cruciferous plants [58,59]. Specifically, P450 is widely involved in the core structure synthesis of glucosinolates. During the synthesis of aliphatic thiosinolates in *A. thaliana*, *CYP79F1* converts short-chain amino acids into long-chain amino acids. *CYP79F2* then oxidizes long-chain amino acids to aldehyde groups. The aldehyde group is activated to an acyl compound by *CYP83A1*. The genes that encode the aldehyde conversion of indole glucosinolates are *CYP79B2* and *CYP79B3*. *CYP81B1* converts the aldehyde group to butyl cyanide, whereas *CYP81F2* and *CYP81F3* participate in the side chain modification of indole glucosinolates. In this study, we analyzed 11 *P450* gene families in seven plant species. These 11 gene families are clustered into *CYP78*, *CYP79*, *CYP81,* and *CYP83* clades. The number of genes varied greatly with species. The *P450* genes of cabbage in the *CYP81* and *CYP78* clades were partly lost due to various evolutionary selection pressures. Comparative genomic analyses revealed that some P450 genes of the *CYP79F*, *CYP81F*, *CYP79B*, *CYP83B*, *CYP83A,* and *CYP78B* families existed only in *Cruciferae*. We speculate that this is related to the fact that glucosinolates are special cruciferous metabolites.

## 5. Conclusions

In this study, 193 glucosinolates biosynthetic genes were identified in cabbage. Some BoGBGs underwent negative selection similar to those in *B. rapa*. RNA-Seq data analysis of six organs in *B. oleracea* and *B. rapa* indicated that many BoGBGs exhibit both organ-specific expression and functional differentiation. Spraying cabbages with five exogenous hormones affected the expression of BoGBGs, and this potentially aids the improvement of the cabbage glucosinolates content by phytohormones application. Overall, this study facilitates unraveling metabolic mechanisms of glucosinolates in *B. oleracea*, providing valuable insights for the development of cabbage varieties with high glucosinolate content through some significative *BoGBG* editing such as *BoCYP83A1*, *BoCYP79F1,* and *BoMAM1*.

## Figures and Tables

**Figure 1 genes-14-00476-f001:**
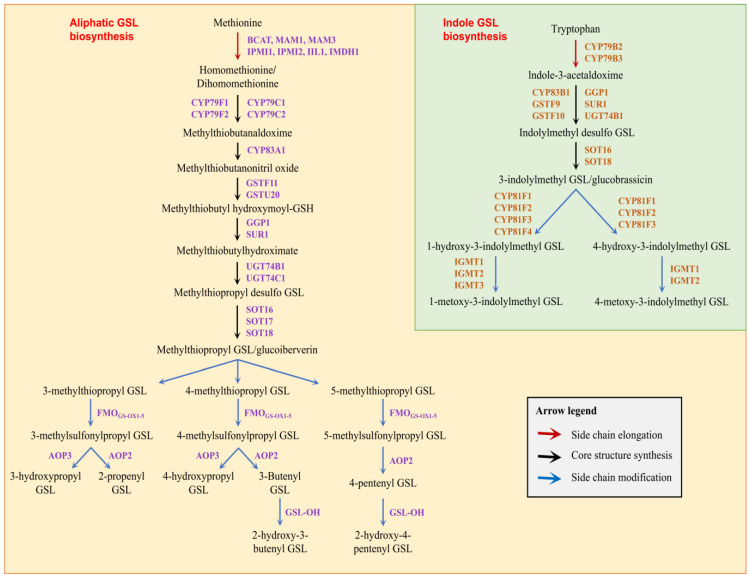
Roadmap for glucosinolate biosynthesis. The compound beside the direction line is the catalytic enzyme, and the compound that the direction line points to is the reaction product.

**Figure 2 genes-14-00476-f002:**
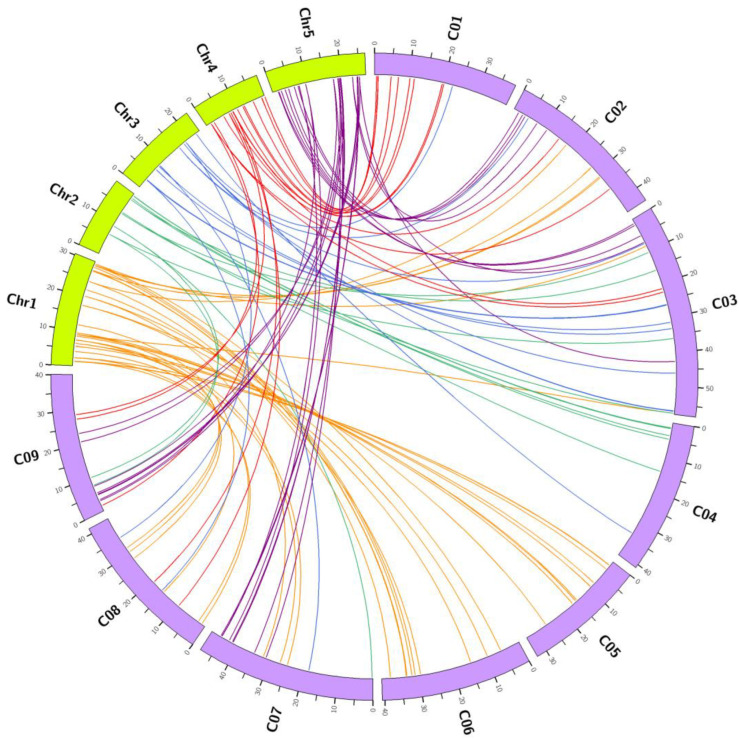
Circos diagram of glucosinolate biosynthetic genes in *Brassica oleracea* and *Arabidopsis thaliana*. C01 to C09 denote *Brassica oleracea* chromosomes, and Chr1 to Chr5 denote *Arabidopsis thaliana* chromosomes.

**Figure 3 genes-14-00476-f003:**
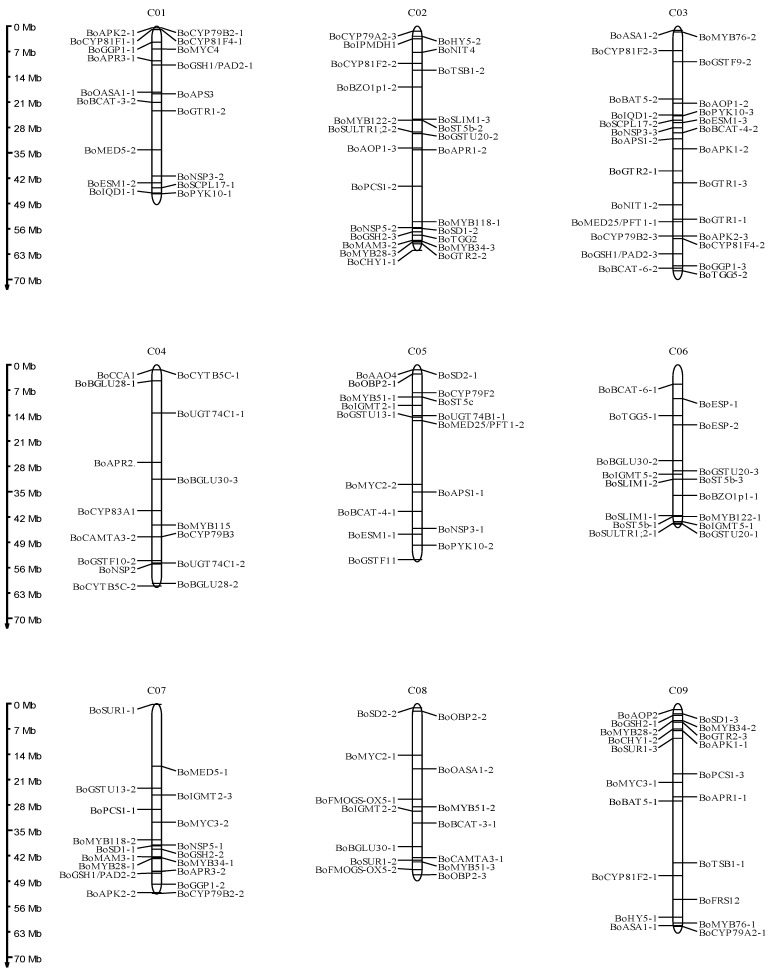
Genomic localization diagrams of the BoGBGs on nine chromosomes of *Brassica oleracea.* C01-C09 denote *Brassica oleracea* chromosomes.

**Figure 4 genes-14-00476-f004:**
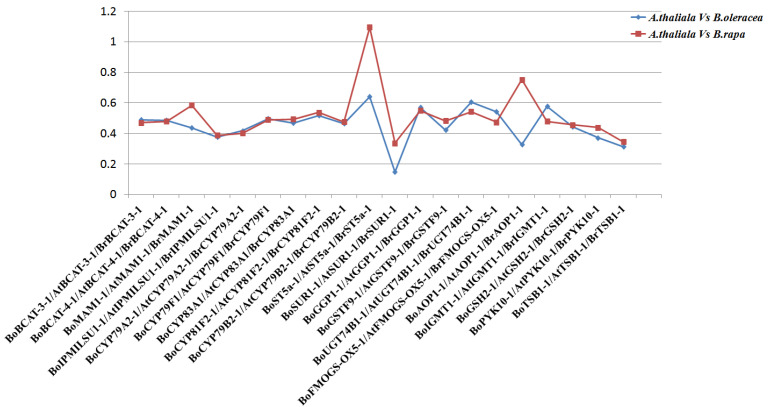
Scatter plot of Ka/Ks values of orthologous gene pairs for GBGs among *Arabidopsis thaliana*, *Brassica oleracea* and *Brassica rapa*.

**Figure 5 genes-14-00476-f005:**
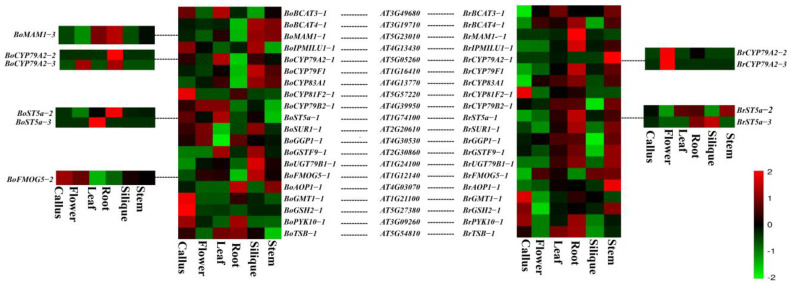
Heat map of homologous gene pairs of glucosinolate biosynthetic genes of *Brassica oleracea* and *Brassica rapa*. The color scale bar represents the FPKM value after the log_2_ transformation.

**Figure 6 genes-14-00476-f006:**
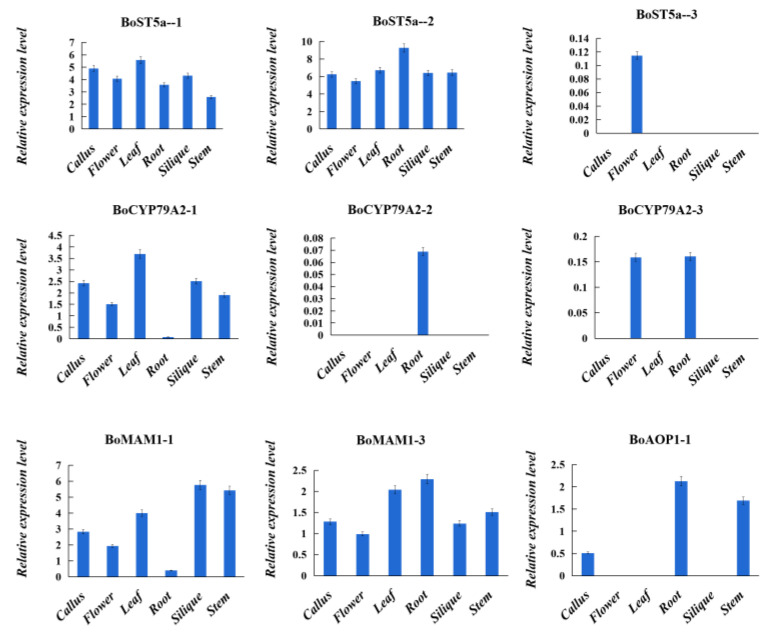
Relative expression levels of nine key glucosinolate biosynthetic genes in different tissues of cabbage.

**Figure 7 genes-14-00476-f007:**
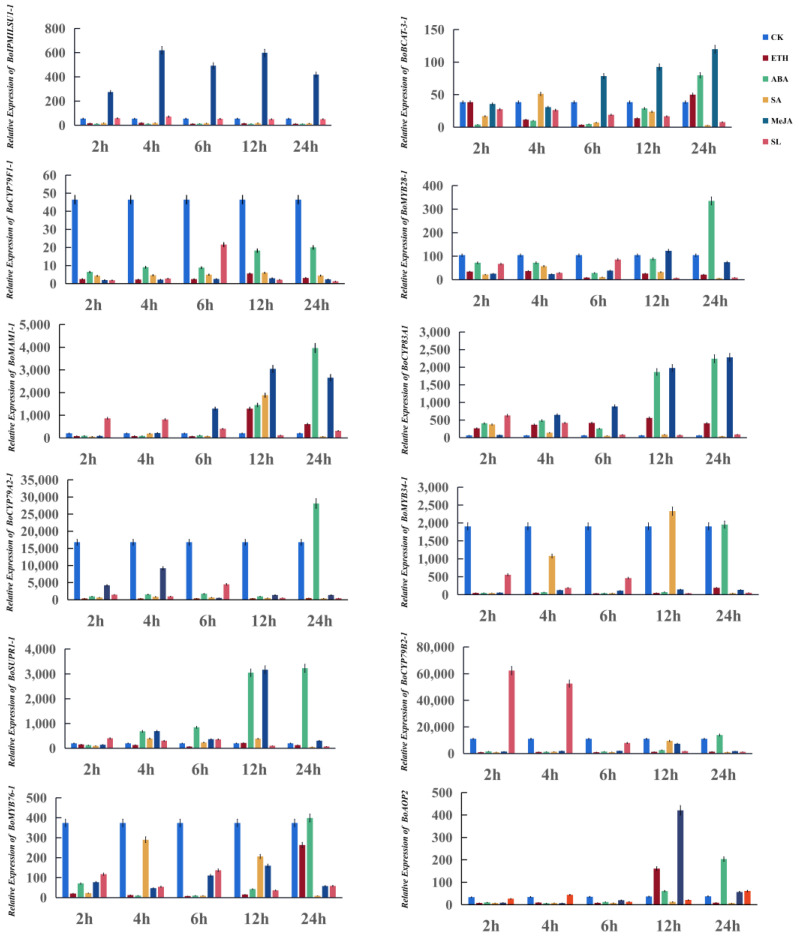
The expression of glucosinolate biosynthetic genes after treatment with exogenous phytohormones. The *Y* and *X* axes represent the relative expression level and time course of exogenous phytohormone treatment, respectively. Leaves were sampled 2, 4, 6, 12, and 24 h after H_2_O(CK), ethylene (ETH), abscisic acid (ABA), salicylic acid (SA), strigolactone (SL), and methyl jasmonate (MeJA) application. The values represent the mean ± SD of three technical replications.

**Figure 8 genes-14-00476-f008:**
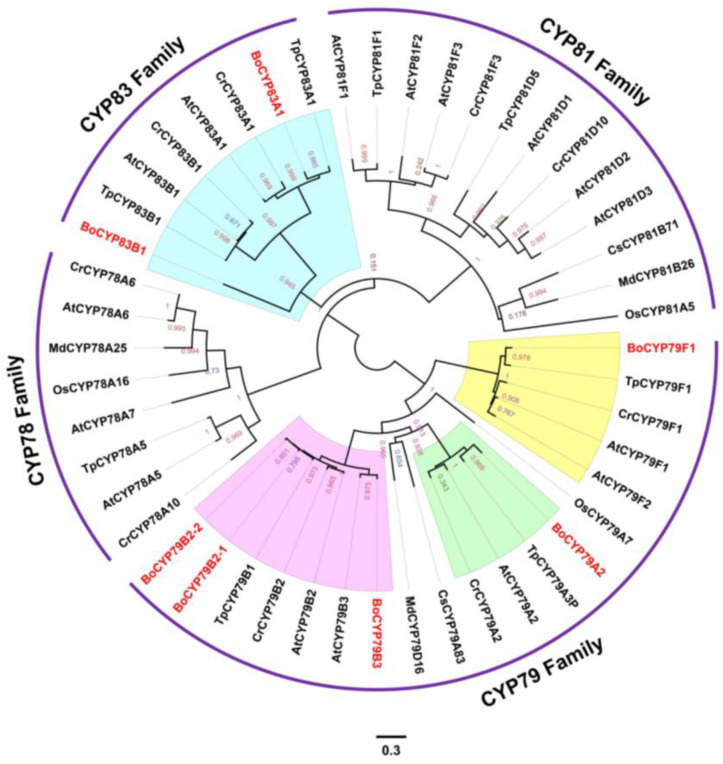
Phylogenetic tree and conserved motif analysis of P450 genes. Protein sequences were aligned by MAFFT (v7.037). The phylogenetic tree was constructed using FastTree with WAG + CAT model. Bo: *Brassica oleracea*; At: *Arabidopsis thaliana*; Tp: *Thellungiella parvula*; Cs: *Cucumis sativus*; Cr: *Capsella rubella*; Md: *Malus domestica*; Os: *Oryza sativa*.

## Data Availability

Not applicable.

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
