# Peer review of "Glucosinolate Biosynthetic Genes of Cabbage: Genome-Wide Identification, Evolution, and Expression Analysis"

_genes, 2023, doi:10.3390/genes14020476_

Round 1
Reviewer 1 Report
Abstract
Line 22-24 Is it possible to clearly specify that MeJA/ETH upregulated/repressed expression of genes responded for side chain elongation, core structure synthesis or side chain modification?
Line 25-27 Described what techniques/means of genetic engineering can be used to regulate or modified GSLs synthesis.
Introduction
Line 34 Regrading to your mention “Due to their effects on….”, what indeed meaning of effects that you would like to present. Please specify description of effect.
Line 38 As sulforaphane is in group of glucosinolates, the relation/link between glucosinolate and sulforaphane should be rephrased.
Add ref. in Figure 1.
Materials and Methods
In topic 2.4, Clarify the cabbages used in this experiment in such a view point of how old of plant, how to spray phytohormone (what device used, volume of spraying, how frequent to spray etc.).
Were whole leaves of cabbage collected to analyze?
What genes used as internal reference/control to validate level of expression.
Discussion
Line 215-216 More discussion about what kinds of selective pressures in meaning. Please present some examples.
Conclusions
Line 281-282 Fill knowledge gap of data obtained in this study with the advantages in further or proposed future work. Wording “through genetic engineering” still not clearly indicate how to take the advantage of data to regulate glucosinolate synthesis.
Reviewer 2 Report
The manuscript entitled, ' Glucosinolate Biosynthetic Genes of Cabbage: Genome-Wide Identification, Evolution, and Expression Analysis' extensively studied the the GBGs in cabbage and studied their expression in different tissues.
1. While studying the transcriptome from public domain, the available datasets of cauliflower and broccoli should also be included as they are also important source of glucosinolates.
2. Tissue specific expression of the selected genes through qRT PCR in cabbage should be included.
3. In the discussion part, significance of the GBGs should be elabotared and future course of action using the present findings should be elucidated.
